Microbes within the building envelope—a case study on the patterns of colonization and potential sampling bias

Davies Lucy R. lucy.r.davies@jyu.fi 1
Barbero-López Aitor 2
Lähteenmäki Veli-Matti 2
Salonen Antti 3
Fedorik Filip 3
Haapala Antti 2
Watts Phillip C. 1
1 Department of Biological and Environmental Science, University of Jyväskylä , Jyväskylä , Finland
2 Department of Chemistry, University of Eastern Finland , Joensuu , Finland
3 Civil Engineering, Faculty of Technology, University of Oulu , Oulu , Finland
Thomas Jonathan
Electronic publication date: 2023 Nov 17
Publication date: 2023
Volume: 11
Electronic Location ID: e16355
Received 2023 Apr 21; Accepted 2023 Oct 4
Copyright: ©2023 Davies et al.
Copyright year: 2023
Copyright holder: Davies et al.
License: This is an open access article distributed under the terms of the Creative Commons Attribution License, which permits unrestricted use, distribution, reproduction and adaptation in any medium and for any purpose provided that it is properly attributed. For attribution, the original author(s), title, publication source (PeerJ) and either DOI or URL of the article must be cited.
License URL: https://creativecommons.org/licenses/by/4.0/

Keywords: Indoor microbiota, Built environment, Occupant health, Building mould

Funding: The Academy of Finland 329883 The work was funded by The Academy of Finland; award no. 329883 to Phillip Watts as part of the CLIHE programme. The funders had no role in study design, data collection and analysis, decision to publish, or preparation of the manuscript.

==============================
Humans are exposed to diverse communities of microbes every day. With more time spent indoors by humans, investigations into the communities of microbes inhabiting occupied spaces have become important to deduce the impacts of these microbes on human health and building health. Studies so far have given considerable insight into the communities of the indoor microbiota humans interact with, but mainly focus on sampling surfaces or indoor dust from filters. Beneath the surfaces though, building envelopes have the potential to contain environments that would support the growth of microbial communities. But due to design choices and distance from ground moisture, for example, the temperature and humidity across a building will vary and cause environmental gradients. These microenvironments could then influence the composition of the microbial communities within the walls. Here we present a case study designed to quantify any patterns in the compositions of fungal and bacterial communities existing in a building envelope and determine some of the key variables, such as cardinal direction, distance from floor or distance from wall joinings, that may influence any microbial community composition variation. By drilling small holes across walls of a house, we extracted microbes onto air filters and conducted amplicon sequencing. We found sampling height (distance from the floor) and cardinal direction the wall was facing caused differences in the diversity of the microbial communities, showing that patterns in the microbial composition will be dependent on sampling location within the building. By sampling beneath the surfaces, our approach provides a more complete picture of the microbial condition of a building environment, with the significant variation in community composition demonstrating a potential sampling bias if multiple sampling locations across a building are not considered. By identifying features of the built environment that promote/retard microbial growth, improvements to building designs can be made to achieve overall healthier occupied spaces.

Introduction

Investigations into microbial communities of occupied spaces of the built environment (the indoor microbiota) typically use methods that collect filtered air particles or dust that accumulate on indoor surfaces (e.g., Maestre et al., 2018; Pekkanen et al., 2018; Fu et al., 2020). These methods facilitate large-scale projects (especially citizen science (Barberán et al., 2015a; Barberán et al., 2015b) and are both straightforward and non-destructive, but these samples cannot separate the microbial contributions of the structural aspects of the building (e.g., the building envelope) from those derived from the building occupants (e.g., hair or skin) (Cao et al., 2021) or the surrounding environment (e.g., soil, pollen grains) (Barberán et al., 2015a), with limited information on how any structural aspects of the building could be contributing to the microbiota of the built environment.

Buildings are complex, three-dimensional structures that contain significant spaces beneath the occupant accessible surfaces. By design, these areas are intended to be dry environments, but humidity can be found in these spaces due to air flow (Fedorik et al., 2021). Evidence of microbial communites inhabiting extreme environments such as hyper-arid areas within the Atacama desert (Schulze-Makuch et al. 2018; Hwang et al., 2021) or the International Space Station (Checinska Sielaff et al., 2019), raises the potential for these building areas to host viable microbes. Some fungi genera and actinomycetes, for example, have been cultured from insulation material samples (Pessi et al., 2002) and can penetrate building structures (Pessi et al., 2002; Airaksinen et al., 2004). The presence and potential contamination of indoor air of these microbes is of significant concern due to their detrimental effects on health (Järvi et al., 2018). Investigating factors influencing microbiota formation, such as building design, environmental conditions and materials, that have an impact on these microbial communities would allow for better predictions and control over building health problems, including material degradation, indoor air quality and human health concerns due to microbial interactions. Establishing a clearer picture and better understanding of the microbes occupying the whole built environment could influence building policies and demonstrate the need to adapt for a healthier home.

Building design and structural condition are therefore likely to be a key determinant of the microbiota composition within building structures due to microenvironmental variation. For example, gradients in temperature, and/or moisture in a building will be an expected consequence of material choice and envelope depth (Fedorik et al., 2021), or cardinal direction in which a structure is facing (Kabátováa & Ďuricaa, 2019). Examining the microbes that live beneath the surface within an occupied space is essential for establishing a fuller picture of the microbiota of the built environment. In-wall sampling though must take these potential gradients into consideration as sampling in just few locations may result in sampling bias.

The goal of this study was to investigate possible differences in microbial (bacteria and fungi) communities, across different locations within a building, that have accumulated over time; thereby highlighting the potential bias that may arise from insufficient in-wall sampling. Additionally, we aimed to identify potential factors driving any variation in microbial communities. To test our approach, we selected a residential dwelling situated within a forested area in Finland. As a stand-alone building with walls facing all cardinal directions, it served as an ideal case-study. Samples were taken by drilling small holes (Ø= 12 mm) into different areas of a home to make a contrast between cardinal directions in which the wall faces, the distance from floor level and the distance from wall joints. Air samples were then extracted from spaces between internal and external walls to obtain a sample of microbes which was then quantified using amplicon sequencing. This novel approach identified significant spatial variation in the bacterial and fungal communities to demonstrate the diversity of microbes within building materials.

Methods

Sampling plan and microbial extraction

On September 20th, 2020, we sampled a traditional Finnish wood-framed house with a natural gravity-driven ventilation located in municipality of Vaala, Finland, that had traditional sawdust and wood shive insulation from 1962 on cast concrete foundation by inserting a sterile tygon tube into 12 mm holes that had been drilled into the walls, with the hole then sealed using modelling clay to prevent collection of ‘indoor air’ from living space. Air from within the building element was pumped (20 min at 3 litres/min) using a SKC universal air pump (model 224-PCMTX4K) over a SureSeal Blank Styrene cassette containing a 25 mm PTFE membrane filter. Holes were drilled in different sampling areas: (1) direction (i.e., the cardinal direction in which the wall is facing), (2) height from floor level (lower, middle and upper) and (3) position (near left wall joining, centre of the wall and near right wall joining) (Fig. 1). Additionally, air samples were taken throughout the day around the outside of the building and from various rooms within the building.

Figure 1 Schematic showing the layout of the sampling site.

Holes were drilled into the walls of a traditional Finnish wood-framed house which contained two bedrooms, a living room and a kitchen. The toilet was located in an out-house. Samples were taken from walls facing different cardinal direction, at different heights (lower, middle and upper), and at different positions (left, centre and right). The grid squares show the different areas sampled. Although only shown on a couple of walls, this scheme was followed on all sampled walls. Sampled walls are highlighted in yellow. Wall temperatures and humidity measurements, where taken, are labelled in white font, outside temperature and humidity in blue, inside in dark-grey, and floor in black. Measurements were taken during the day of sampling.

DNA extraction and sequencing

Filters were soaked in molecular grade water for 12 h. This was to ensure microbes were washed ‘out’ of the filters and easily accessible for DNA extraction. DNA was extracted from the filter and its water using Qiagen DNeasy® PowerSoil® Pro Kit according to the manufacturer’s protocol, but with the following adjustments: (1) PowerBead Pro Tubes were vortexed for 20 min at maximum rpm speed. DNA was also extracted from control samples: only buffer, molecular grade water and buffer, and unused filters and buffer.

Bacterial and fungal taxa was then identified by amplicon sequencing performed on an Illumina NovoSeq by NovoGene Ltd (https://www.novogene.com/us-en/). Bacterial 16S V4 region (universal primers GTGCCAGCMGCCGCGGTAA and GGACTACHVGGGTWTCTAAT) and the fungal ITS2 region (universal primers CATCGATGAAGAACGCAGC, TCCTCCGCTTATTGATATGC) were targeted.

Processing and analysing fungal and bacterial sequences

Primers were removed using cutadapt v1.10 (Marcel, 2011) on ITS reads, while DADA2 v1.18 (Callahan et al., 2016) was used to remove primers and truncate (forward reads at base 220, reverse reads at base 200) the 16S reads. DADA2 was used for both ITS and 16S data to merge paired-end reads, remove chimaeras and identify amplicon sequence variants (ASVs) (Callahan et al., 2016). Taxonomy was assigned to ITS and 16S ASVs using UNITE v. 10.05.2021 (Nilsson et al., 2015) and the SILVA v.138 database (Quast et al., 2013) respectively. decontam v.1.14.0 (Davis et al., 2018) was used to eliminate potential contaminant ASVs identified in the control samples, using the prevalence method and the probability that a read is a contaminant threshold of 0.5 (Davis et al., 2018).To remove low frequency ASVs which were most likely a result of sequencing errors, while also avoiding the removal of rare ASVs, ASVs that had a total count of 20 across the entire dataset were removed.

Statistical analysis

Statistical analysis was performed using R v.4.1.3 (R Core Team, 2021). The package FEAST v.1.0 (Fast Expectation-mAximization microbial Source Tracking) (Shenhav et al., 2019) was used to estimate the contributions of the microbial communities from indoor air and outdoor air to the microbial communities of the in-wall samples. To do this, the air samples were labelled as ‘sources’, and the wall samples were as labelled ‘sinks’. Total Sum Scaling (TSS) normalization was used to remove any bias related to differences in sequencing depth in different libraries by dividing each ASV count with the total library size per sample. Analyses of the microbial communities was then performed on the relative abundance of each ASV using phyloseq v.1.38.0 (McMurdie & Holmes, 2013) and the package vegan v.2.5-7 (Oksanen et al., 2019). The package vegan was used to calculate the alpha diversity measures (the number of different taxa groups and their abundance, and the number of distinct taxa) and beta diversity measures (the diversity differences between two samples). Predictors of variation in alpha diversity was assessed using a generalized linear model that contained alpha diversity as the response variable and direction, height and position as predictor variables. We detected no over-dispersion in the model. We compared the full model with reduced models from which a predictor variable was omitted using a F-test to obtain p-values. The package vegan was used to conduct PERMANOVA tests (using the adonis2() command, using the Bray Curtis distance method, and setting permutations to 999) to assess predictors of variation in beta diversity, and was also used to conduct a distance-based redundancy analysis (dbRDA) ordination method which we used to visualise differences between the microbial communities across different groups (Legendre & Anderson, 1999). To examine the ASVs driving any variation in the beta diversity, the SIMPER() command from vegan was used to calculate similarity percentages (Clarke, 1993). From the identified genera, a Kruskal–Wallis test was conducted, followed by a Dunn test to assess pair-wise significant differences (Dinno, 2017). Statistical significance was based on adjusted p-values using the Bonferroni method (Dunn, 1961). Plots were produced using ggplot2 v.3.3.6 (Wickham, 2016) and patchwork v.1.1.2 (Pedersen, 2022).

Results and Discussion

To quantify patterns of microbial colonization, we first measured alpha diversity using Shannon diversity (the number of different taxa groups and their abundance), and amplicon sequence variant (ASV) richness (the number of distinct taxa). For fungal communities, the height at which the samples were taken had a significant effect on Shannon diversity (Table 1), with the lowest diversity found in the middle of the wall (although a Tukey posthoc test did not show pairwise significant differences: Lower vs Middle Padjusted = 0.24; Middle vs Upper Padjusted = 0.12; Lower vs Upper Padjusted = 0.96) (Fig. 2B). There were no significant differences between the cardinal direction of the wall and positions (Figs. 2A, 2C; Table 1). Although not significant (Table 1), height from ground showed a qualitative pattern where the alpha diversity measurements of the bacterial communities decreased from the lower to the upper part of the wall (Fig. 3B). We were therefore interested in examining whether the Shannon diversity and richness correlated with a quantitative measurement of the height of the wall. When taking the distance from the floor whereby the holes were drilled into consideration as linear variables (20 cm, 120 cm, 220 cm), we found the Shannon diversity of bacterial communities significantly decreased as the distance from the floor increases (F1,21 = 4.12, P < 0.04). As with the alpha diversity of fungal communities, there were no significant differences between the directions and positions (Figs. 3A, 3C; Table 1).

Table 1 Statistical results from the F-tests for the alpha diversity of fungal and bacteria communities.

Results were obtained by comparing a generalized linear model containing either Shannon diversity or number of individual ASVs as the response variable and direction, height and position as the predictor variables with a reduced model where a predictor variable was omitted.

 	 	Shannon diversity index	Number of individual ASVs	
Kingdom	Variable	Estimate ± s.e.	R-squared	F d.f	P-value	Estimate ± s.e.	R-squared	F d.f	P-value	
 	Direction	2.06 ± 0.39	0.13	1.841,21	0.16	21.5 ± 10.34	0.28	2.091,21	0.12	
Fungi	Height	2.15 ± 0.27	0.13	4.111,21	0.03*	29.64 ± 8.59	0.01	0.381,21	0.69	
 	Position	2.04 ± 0.25	0.04	0.371,21	0.68	35.71 ± 7.23	0.11	1.671,21	0.21	
 	Direction	3.05 ± 0.26	0.13	1.761,21	0.18	202.67 ± 38.55	0.18	2.501,21	0.08	
Bacteria	Height	3.31 ± 0.18	0.15	2.881,21	0.08	236.73 ± 28.54	0.11	2.841,21	0.08	
 	Position	2.93 ± 0.18	0.02	0.991,21	0.38	182.93 ± 26.28	0.03	1.571,21	0.23	
Notes.

* Significant P-value.

Figure 2 ITS alpha diversity analysis: The average Shannon diversity index and number of individual ASVs (±standard error of mean).

Comparing (A) direction, (B) height, (C) position. Smaller points represent raw data from each sample. Sample sizes: internal = 5, north = 7, west = 6, south = 4, east = 6; lower = 11, middle = 9, upper = 8; left = 8, centre = 14, right = 6. An * indicates a significant F-test result.

Figure 3 16S alpha diversity analysis: The average Shannon diversity index and number of individual ASVs (±standard error of mean).

Comparing (A) direction, (B) position, (C) height. Smaller points represent raw data from each sample. Sample sizes: internal = 5, north = 7, west = 6, south = 4, east = 6; lower = 11, middle = 9, upper = 8; left = 8, centre = 14, right = 6.

Fungi and bacteria can inhabit diverse environments (Storze & Hengge, 2011; Haruta & Kanno, 2015), with the community composition dependent on the outcome of selection and competition among taxa. In natural environments, environmental variation in, for example, humidity (Wang et al., 2021), moisture (Borowik & Wyszkowska, 2016), temperature (Nottingham et al., 2022), and pH (Scholier et al., 2022) are important drivers of fungal and bacterial composition and activity. Just as there are environmental gradients in nature (Wang et al., 2021), typically, increasing higher up you go in a wall is associated with a dryer and warmer environment (Fedorik et al., 2021). It is these variations that would elicit changes in microbiota composition.

For fungal and bacterial richness, the variable that had the greatest variance was the cardinal direction in which the sampled wall was facing (Table 1). Samples taken from west facing walls had the highest number of individual ASVs (Figs. 2B, 3A). Previous studies on indoor microbiota have also shown cardinal direction to have an important impact on bacterial communities (Fahimipour et al., 2018; Horve et al., 2020). Though focusing on viable bacteria on surfaces, rather than in-wall sampling, Horve et al. (2020) found west facing rooms with windows to have a higher abundance of viable bacteria compared to rooms facing other cardinal directions due to direct sunlight (Horve et al., 2020). The intensity of direct sunlight will vary across different seasons and as such, studies have shown exposure to indoor microbes can vary across seasons (Garrett et al., 1997; Rintala et al., 2008). The heat generated by direct sunlight can increase indoor humidity - a factor that contributes to these seasonal variations (Frankel et al., 2012; Knudsen, Gunnarsen & Madsen, 2017). Therefore, when investigating microbial compositions, humidity within the building envelope would likely be an important environmental factor causing variation.

To examine the most relatively abundant taxa, we grouped the ASVs at genus level and identified the top fifteen (see Figs. S1 and S2 for fungi and bacteria, respectively). Several of the top relatively abundant genera identified in this study are associated with building and occupant health. For example, fungal species of Antrodia, and Heterobasidion are key house-rot fungi, as they decay wooden building materials (Huckfeldt & Schmidt, 2006; Schmidt, 2007; Schmidt & Huckfeldt, 2011; Gabriel & Švec, 2017; Haas et al., 2019). Sarocladium species are associated with problems in biodegradation of mineral-based materials (Ponizovskaya et al., 2019). Species of Aspergillus are reported from multiple studies of the indoor environments (Tanaka-Kagawa et al., 2005; Mousavi et al., 2016; Chen et al., 2017) and some species from this genus, along with species of Phialocephala, affect the severity of asthmatic symptoms (Hedayati, Mayahi & Denning, 2010; Dannemiller et al., 2016; Mousavi et al., 2016). Many of the top relatively abundant bacterial genera have also been identified in studies on the indoor environment. Acinetobacter (Hui et al., 2019; Wu et al., 2022), Cutibacterium (Sun et al., 2022), Staphylococcus (Moon, Huh & Jeong, 2014; Madsen et al., 2018) and Blaudia (Fu et al., 2021), for example, occur in samples of indoor dust and swabs. Previous studies have also made links between these genera and negative impacts on human health (Kozajda, Jeżak & Kapsa, 2019; Fu et al., 2021; Sun et al., 2022; Wu et al., 2022). Further investigations are necessary to determine if similar risks exist when these genera are inhabiting areas that are not frequently in direct contact with humans. Regarding fungal genera, it is possible that they may remain dormant and only become problematic when certain environmental changes occur, such as excess moisture. But identifying the presence of these microbial genera is important because it provides valuable insights into the potential risks and could allow for mitigation of problems.

As the microbial communities observed in household surfaces can be sourced from building occupants (Cao et al., 2021), or the surrounding environment (Barberán et al., 2015a) and geographic location (Chen et al., 2017), we wanted to examine the potential impact of the indoor and outdoor microbial communities on the communities found within the walls. Source tracking revealed that the fungal communities found within the wall are likely to be independent of the communities found in the indoor and outdoor air (Fig. S3). Within the bacterial communities, half of the wall samples had more than 50% of ASVs that could not be sourced to the air samples (Fig. S4), while the remaining had more of a mix of air and unknown sources (Fig. S4). Of the top fifteen bacterial genera (Fig. S4), a number of these could be traced to human as a source, such as Faecalbacterium (Bai et al., 2023), Blaudia (Dobay et al., 2019) and Cutibacterium (Sun et al., 2022), or could be traced to soil, for example Sphingomonas (White, Sutton & Ringelberg, 1996) and Acinetobacter (Hui et al., 2019). Understanding when microbes could colonize the building, such as during material processing or when being stored on a building site, and investigating if their relative abundance increases over time, would provide information on their potential sources and dynamics within the built environment. This knowledge could contribute to a better understanding of the factors influencing the microbiota of a building. Additionally, as many microbes have dormant stages, it would be interesting to determine which of these taxa are viable.

We next quantified patterns in beta diversity (the diversity between two microbial communities). To determine the sampling variable most influencing differences in communities, we analysed Brays-Curtis dissimilarity (index based on the abundance of individual ASV groups) and Jaccard distance (index based on the presence and absence of individual ASV groups). For both measures, direction particularly caused variation across the fungal and bacterial communities (Table 2, Figs. 4A, 5B). As with height, differences in macroenvironment will influence the survival and selection of different microbial taxa (Storze & Hengge, 2011; Haruta & Kanno, 2015). Variation among walls in their exposure to wind and/or sun can generate differences in envelope temperature that could explain variation in the community composition.

Table 2 Statistical results from the ADONIS tests for the beta diversity of fungal and bacteria communities.

Distance matrices Brays–Curtis dissimilarity and Jaccard were both tested with each variable used as the predictor variable.

 	 	Brays–Curtis	Jaccard	
Kingdom	Variable	R-squared	P-value	R-squared	P-value	
 	Direction	0.16	0.31	0.15	0.35	
Fungi	Height	0.08	0.71	0.07	0.68	
 	Position	0.07	0.67	0.07	0.74	
 	Direction	0.21	0.08	0.19	0.06	
Bacteria	Height	0.03	0.90	0.04	0.97	
 	Position	0.08	0.25	0.07	0.41	

Figure 4 Fungi beta diversity analysis.

The ordination was obtained by conducting a distance-based redundancy analysis based on Brays–Curtis dissimilarity matrices for (A) direction, (B) height, (C) position. Each smaller point represents the fungal community in a sample. Ellipses represent a 95% CI centred around a centroid shown by a transparent, larger point. Sample sizes: internal = 5, north = 7, west = 6, south = 4, east = 6; lower = 11, middle = 9, upper = 8; left = 8, centre = 14, right = 6.

Figure 5 Bacteria beta diversity analysis.

The ordination was obtained by conducting a distance-based redundancy analysis based on Brays–Curtis dissimilarity matrices for (A) direction, (B) height, (C) position. Each smaller point represents the bacterial community in a sample. Ellipses represent a 95% CI centred around a centroid shown by a transparent, larger point. Sample sizes: internal = 5, north = 7, west = 6, south = 4, east = 6; lower = 11, middle = 9, upper = 8; left = 8, centre = 14, right = 6.

To investigate how the most abundant ASVs contributed to the variability, we examined the differential abundance patterns of the top fifteen genera across the cardinal directions (see Figs. S1 and S2 for fungi and bacteria, respectively). To do this, we used similarity percentages (Clarke, 1993) and identified the genera that most contributed to the beta-diversity measures based on Bray–Curtis dissimilarity indices. Among the top fifteen relatively abundant genera, eleven fungal and ten bacterial genera were present across the pairwise comparisons and were therefore considered to be influencing the observed variations (Table S1). A Kruskal–Wallis and subsequent Dunn test on the eleven fungal genera found a significant difference in the relative abundance of Ceriporiopsis; the relative abundance was significantly higher in west-facing walls compared to east (Table S2). The relative abundance of the bacterial genus Bradyrhizobium was significantly higher in internal walls compared to north-facing walls (Table S2). A limitation with our data, though, is sample size, and a larger dataset would have greater statistical power. We would expect buildings with different uses and built using different materials to exhibit different patterns across the structure. Further research involving a diverse range of building types and materials would provide an interesting and comprehensive understanding of the factors driving compositional changes in building microbiota. Additionally, a further direction could be to also quantify the microbial load at different locations, although this would be challenging to do so (Galazzo et al., 2020). Nonetheless, despite being limited to one building, we show the occurrence of compositional changes in microbiota, even across a relatively small building. As such, our data shows how in-wall sampling needs to encompass multiple locations within and across different regions of any building element to avoid sampling bias when studying the building microbiota.

Conclusion

Here we present a case study using an overlooked approach in sampling and understanding the processes that determine the composition of the building microbiota. Our aim was to characterize the microbial communities within the building envelope and determine which building elements, if any, have the largest impact on compositional variation found within structures. Our results show significant differences in the alpha diversity across different heights of the wall and that factors such as cardinal direction can elicit variation in the community composition. While our study focused on one building, we show the potential for diverse microbiota across the building envelope and that to get a fuller picture of the microbiota of the built environment, variation across height or the cardinal direction of the building must be taken into account during sampling. The identification of independent communities within the walls shows that future investigations should therefore think of a building as its own ecosystem amongst the indoor biome (Adams et al., 2015). Studies should take into consideration these ‘hidden’ microbial communities that have the potential to cause damage to buildings and cause problems to occupants’ health while also accounting for microenvironmental changes across building structures. These preliminary findings serve as the foundation for expanding our approach and delving deeper into investigating the microbiotas beneath the surface.

Supplemental Information

Supplemental Information 1 Heatmap showing the top 15 relatively abundant fungi

Darker colour represents higher relative abundance in percentage.

Click here for additional data file.

Supplemental Information 2 Heatmap showing the top 15 relatively abundant bacterial genera

Darker colour represents higher relative abundance in percentage.

Click here for additional data file.

Supplemental Information 3 Pie charts showing the potential sources of fungal ASVs in each wall sample

Sources were labelled as indoor air and outdoor air. Where ASVs didn’t fall in to these categories, they were labeled as coming from an unknown source.

Click here for additional data file.

Supplemental Information 4 Pie charts showing the potential sources of bacterial ASVs in each wall sample

Sources were labelled as indoor air and outdoor air. Where ASVs didn’t fall in to these categories, they were labeled as coming from an unknown source.

Click here for additional data file.

Supplemental Information 5 SIMPER output

Lists of genera driving the composition variation in each comparison of cardinal directions.

Click here for additional data file.

Supplemental Information 6 Dunn test results

Statistical tables showing the results of the Kruskal–Wallis tests, and the differences between each cardinal direction. Genera identified in the SIMPER test were analysed.

Click here for additional data file.

We would like to thank Pekka Ylimäki for his assistance in sampling the cottage and Ilze Brila for her valuable guidance on the analysis. The authors would also like to thank the reviewers for their time. We sincerely appreciate their valuable comments and expertise, which helped improve the quality of the manuscript.

Additional Information and Declarations

Competing Interests

Author Contributions

Data Availability

The authors declare there are no competing interests.

Lucy R. Davies conceived and designed the experiments, performed the experiments, analyzed the data, prepared figures and/or tables, authored or reviewed drafts of the article, and approved the final draft.

Aitor Barbero-López conceived and designed the experiments, performed the experiments, analyzed the data, authored or reviewed drafts of the article, and approved the final draft.

Veli-Matti Lähteenmäki conceived and designed the experiments, performed the experiments, analyzed the data, authored or reviewed drafts of the article, and approved the final draft.

Antti Salonen analyzed the data, authored or reviewed drafts of the article, and approved the final draft.

Filip Fedorik conceived and designed the experiments, performed the experiments, analyzed the data, authored or reviewed drafts of the article, and approved the final draft.

Antti Haapala conceived and designed the experiments, performed the experiments, analyzed the data, authored or reviewed drafts of the article, and approved the final draft.

Phillip C. Watts conceived and designed the experiments, performed the experiments, analyzed the data, authored or reviewed drafts of the article, and approved the final draft.

The following information was supplied regarding data availability:

The 16S and ITS raw sequences are available at GenBank: PRJNA958164.

The metadata is available at figshare: Davies, Lucy (2023). Metadata. figshare. Dataset. https://doi.org/10.6084/m9.figshare.22059653.v2.

The code for analysis is available at figshare: Davies, Lucy (2023). R script for analysis. figshare. Online resource. https://doi.org/10.6084/m9.figshare.22059686.v2.

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
