# Peer review of "Microbes within the building envelope—a case study on the patterns of colonization and potential sampling bias"

_PeerJ, doi:10.7717/peerj.16355_

## Round 0.1 · original submission · Major Revisions

We have obtained three high-quality reviews of your interesting work. Although all reviewers consider that your work is interesting, they all mention the need for a better discussion of the observed taxa, regarding their distribution, co-occurrence, etc. Another concern mentioned often is the limitation incurred due to you only having studied a single dwelling. Please address these and the other issues.

·

Basic reporting

The submitted article presents a study of the airborne microbiome contained in the buiding envelope, in order to highlight variables which may drive the microbial community structure and composition.

The article is clear and comprehensible; however, few points should be more detailed:

- Line 20 : The notion of “distance from vegetation” is not sufficiently clear or explicit.

- Line 51 to 55: References are needed for this part of the state of art. In addition, the health problems associated to microbial growth on building materials or indoor environment should be deepened in order to improve understanding of the issues linked to theses microorganisms.
It is indeed necessary to insist more on the issues related to this study, because its importance is not sufficiently clear from the introduction. In addition, the way the microorganisms present between walls may impact the indoor environment or building structure is not really highlight.

There are no English language problems or the structure of the article on the whole document, just some minor typo errors:

- Line 38 : “i.e” needed at the start of the explanation in brackets.

- Line 41 : remove “for” between “facilitate” and “large-scale”.

- Line 48 : add a comma after “environments”.

Figures and tables are useful, but the authors should show the significance of the tests performed on the alpha diversity graphics directly on the figures to facilitate the reading and the conclusions drawn.

Experimental design

Although the method used to study the microbiome contained in the walls of the building is well described, whether it is at the level of the sampling methods, the sequencing of genetic markers or the processing of these data, the study presents certain limits in its analysis.

i) No quantification had been presented, such as bacterial and fungal quantification by qPCR technique. Therefore, it is impossible to determine whether the differences observed on the different communities are solely due to the factors studied but not to differences in population size. It is then complicated to analyze and compare in a sufficiently precise way the various samples obtained without having this information.

ii) Apart from the sampling of the house envelope, there was no sampling of indoor or outdoor air. In air microbiome studies, it is important to have these samples as a reference in order to know how the diversity and composition of the microbial communities in the microenvironments studied are affected in relation to the surrounding environment.

iii) Were rarefaction curves and normalizations of the number of reads per sample performed on this data set? This is not indicated in the materials and methods section. If this has not been done, it may lead to important biases in the analysis when comparing communities with different sequencing depths.

iv) No values for the environmental parameters are presented, either for the indoor environment, the indoor air or the sampled air located between the walls. This information is essential to distinguish or link the effects of the observed factors from those of the ambient environmental conditions.

v) Finally, the study of beta biodiversity enables the observation of the different criteria that can affect the diversity and composition of the microorganisms contained within the walls of the house. It could have been interesting to look specifically at which microbial taxa could be affected by the different factors highlighted, and if correlations could also be drawn in terms of taxonomic assignments.


Also, just one methodological point to clarify:

- Line 94 to 98: Does theses primers were designed during this study or had previously been used in previous works?

Validity of the findings

Setting aside the previous points raised about the experimental design, the study presents data that are interesting. However, one of my main concern about this study is that only one building had been analyzed. The lack of replicate concerning the sampled houses induce a dataset too small to drive any significant pattern. The authors should perform their analysis on at least 3 different buildings in order to validate the repeatability of their observations and conclusions.

Additional comments

Overall, the topic presented in this study is very interesting. The microbiome contains within the building materials and inside the building envelope is rarely considered, and the objectives of the study cover a really needed spot in order to better understand the building ecosystem as a whole and the sampling biais associated to its characterization.

Unfortunately, the experimental design presents too many flaws which complicated the robustness of the conclusion drawn. The lack of data concerning for example the size of the microbial communities sampled, the environmental conditions, or the number of sampled building lead to insufficient experimental data to support the significance of the findings of this study.

The authors should continue their work on this topic because of its relevance, with a more substantial study, larger datasets and taking into account more technical and environmental parameters.

Reviewer 2 ·

Basic reporting

Davies et al. present a study on the bacterial and fungal communities within the walls of a traditional Finnish house, collected via aerosol sampling. The authors describe modest but significant differences in the community diversity associated with sampling height, while the other sampling parameters did not significantly influence community diversity. Additionally, none of the sampling parameters were significantly associated with differences in community composition. The authors robustly analyzed and nicely described the diversity differences, and it was good to see potential contamination addressed. Overall, this work provides helpful insights about parameters to consider when sampling the built environment. With further description of the community composition, I would recommend the manuscript for publication.

Experimental design

The analysis feels incomplete without addressing the observed taxonomic composition of the community. What taxa were observed and in what proportions? Were any taxa differentially abundant across the sampling parameters tested? This is the main item I see as missing from the manuscript.

The provided analytical scripts are sensible and well documented, and the bioinformatics method appear to be robust.

It is unclear how generalizable these data are, both because of the unique characteristics of the sampled structure and that fact that only one building was sampled. For example, Chase et al. found that geography significantly shaped the microbiome of office spaces (https://journals.asm.org/doi/10.1128/mSystems.00022-16). Seasonality was also found to have a significant effect; the authors should at least report when their samples were collected (Sept 2020, based on NCBI BioSample metadata).

Measurements of temperature, material moisture content, etc associated with the sampled locations would have been nice to support the authors claims about micro-environments associated with the sampling parameters.

Do the authors suspect that there are differences in biomass associated with the sampling parameters? Ideally a method like qPCR could be used to quantify this, although given equivalent air volumes filtered for each sample, the recovered DNA concentrations can be a rather rough proxy.

Validity of the findings

If the authors have DNA concentration values, I would like to see them use these in their Decontam analysis, employing the frequency method. It would also be good to see PERMANOVA results comparing pre- and post-decontamination samples to the negative control samples.

Additional comments

Are aerosol samples the most appropriate for this environment where there is likely little airflow? The authors contend that this is a good relatively non-invasive sampling method, however I would be hesitant to make such a claim without comparison to potentially more invasive samples swabbed from interior wall surfaces.

I would have liked to see comparison to the communities present in the occupied space of the house, and to a lesser degree corresponding locations on the exterior of the structure. Especially as the authors make the claim that microbes in the interior wall space could have implications for human health, it would be good to show that these microbes are also found in the living space.

It was not immediately clear to me what was meant by ‘internal’ for the direction parameter, although I assume this corresponds to walls between two interior spaces.

Reviewer 3 ·

Basic reporting

I found this to be a clear and concise paper that was interesting to read.

However, at times, the discussion feels superficial, which in part relates to my comments in the experimental design section. For instance, at lines 149-150, there is an opportunity to further expand on the trend presented and connect it back to their findings. There are two major concepts of the paper that I believe could be better connected to existing research:
1. The discussion (or introduction) could be improved by acknowledging that other studies have carefully surveyed individual buildings to understand the effect of space before.
2. The idea of treating buildings as ecosystems is not new and some other literature taking this mindset should be acknowledged.

I found a couple of minor grammatical errors, including the following:
1. line 63: I found the phrasing "beyond the surface" confusing and potentially misleading.
2. line 67: I would recommend deleting "what are" so the sentence just reads: "...to determine the main...".
3. line 77: I would delete "a" before "natural gravity-driven".
4. line 78: I would add "the" before "municipality".
5. line 102: There is an extra "on" after the Callahan et al. citation.
6. line 103: I found the phrasing "from the 16S data" confusing. I would recommend rephrasing.
7. line 122: Should p-value be lowercase here?
8. line 123: There is a missing "was" after "vegan".

I also had one recommendations regarding the figures' presentation:
1. Figure 1: In the caption or on the figure, would it be possible to indicate which walls were sampled in some way (particularly which interior ones)?

Experimental design

The research question was well defined, relevant, meaningful, and in the scope of PeerJ. The methods were described in sufficient detail to be able to replicate the study.

I have two significant methodological concerns with the paper:
1. The study only covered one house, but provides a useful case study and suggests the need for future additional studies that cover more homes and can address whether the effects are specific to this home or indicative of patterns found elsewhere. I recommend that the authors more clearly acknowledge this limitation in their results and discussion. Other studies have demonstrated the incredible variability possible between homes, even within a city/climate region (and single home studies have previously been useful to examine in depth a specific question).
2. The study did not take any humidity readings or otherwise characterize the environment within the walls that was sampled. In the abstract, they mentioned that one of the motivations for their study was to "determine the variables that influence any patterns" (lines 24-25). As humidity and other environmental conditions are known to have strong effects on microbial communities, I recommend that the authors both (1) modify the phrasing at lines 24-25 to specifically indicate the variables tested here and (2) acknowledge more fully how humidity and other parameters may drive the community (e.g., by expanding at line 167). Related to this, at line 47-48, the authors mention that "these areas are intended to be dry environments" and then detail how even if this is the case microbes may still be growing. I would recommend that the authors also acknowledge that depending on different parameters (the location of the home, any leaks, etc), there may be significant humidity in these spaces.

Another significant concern of mine is that the authors did not present any discussion of the specific taxa found by their sampling. I highly recommend that the authors consider expanding their discussion to consider specifically which fungal and bacterial ASVs are found at each sampling location. Were there any common taxa found at each site? Are there taxonomic differences based on direction, height, position? One of the points made in the introduction is that by air sampling within a building, we're missing these envelope microbes. Are these envelope microbes different from the ones found in Finnish homes previously surveyed in the literature?

I also had a minor confusion at line 110-111 ("ASVs that appeared less than 20 times across all of the data were also removed."). Is this based on whether an ASV was found in less than 20 samples or related to the read depth? This confusion needs to be addressed because the choice would change how I interpret these results.

Validity of the findings

I found the overall conclusion of the paper (that it is important to account for microbial communities within a building's envelope as a potential source of variability) to be supported by the study design. As I described above in the experimental design section, there were a couple of facets of the study design that limit generalizability and a missing opportunity to more fully explore the question of their study. And, as noted in the basic reporting section, there are areas where the findings could have been more fully discussed/put in context.

Additional comments

I have some other comments on this paper:
1. I highly recommend that the authors rephrase their title to better reflect their study. At the very least, the title should mention that they only considered one building and "and potential sampling bias" could be rephrased to more clearly reflect why it is mentioned. A minor point is that I would also delete the "the" before "building elements."
2. line 18: I recommend adding "or filters" to mirror the introduction where both were mentioned as important.
3. line 44-45 ("and not from structural aspects of the building"): I would rephrase to temper this statement by the authors. While the degree of influence of microbes growing within walls of a structure, etc on communities collected by air filters or swabbing surfaces may be unknown, it is likely that they are still contributing to the indoor microbiome. The current wording feels like it is overstating the lack of influence to make the case for their study without evidence to justify this statement. Despite this, I do believe this study was still needed to encourage study in this area; and it lays groundwork for future work to assess the degree and ways that microbial communities within walls/etc influence the indoor microbiome (and possible implications on human health).
4. lines 151-159: These two paragraphs seemed to overlap and I believe they could be combined to be more straightforward (maybe still in two paragraphs). My confusion in part stemmed from the use of "variance" at line 151. Specifically, I wasn't sure about the difference between "variance" and "beta diversity" mentioned in the next paragraph because both cite Table 2. Maybe at line 152, Table 1 was supposed to be referenced? If so, then that would make the difference between the paragraphs more clear to me and may resolve this comment.

---

## Round 0.2 · Minor Revisions

The original Academic Editor is not available so I have taken over handling this submission.

While the authors have addressed the majority of the concerns raised by reviewers in the first review, all three reviewers still had some minor concerns that need to be addressed, with two of the reviewers highlighting the title of the manuscript. I also particularly agree with reviewer one's comments about the rarefaction and normalization of the data and the need to explain how this has been taken into account (there are many reviews around this area).

·

Basic reporting

The authors have taken into account most of the previous comments and have provided constructive responses to the issues raised. I thank the authors for all the added information and the development of several part of the manuscript. There are still few points which deserve to be clarified.

Experimental design

Concerning the previous limits about the experimental design, the authors have discussed the different points by adding substantial contents and references. The addition of results and data about the outdoor and indoor microbial communities is indeed really interesting and highlights new perspectives for this work.
But two point needs to be detailed further.
• Concerning the issue with the absence of quantitative data and the use of qPCR, may be my question was not very clear. The goal to use qPCR alongside NGS sequencing is not to quantify the composition of the main microbial taxa, but to quantify the total bacterial or fungal DNA and provide an useful comparison point to assess the potential differences between samples. This method is not exempt from bias and should not be directly correlated to the diversity obtained by NGS sequencing, but offers a new look to analyze the dataset.
• My question about the rarefaction curves and normalizations of the number of reads per sample have not been answer. I have well understand that the analyses were performed on relative abundance of each ASV for each sample. It is still unclear how the gap between the size of the different sample libraries has been handle. As mentioned in several studies such as :
i) Cameron, E.S., Schmidt, P.J., Tremblay, B.JM. et al. Enhancing diversity analysis by repeatedly rarefying next generation sequencing data describing microbial communities. Sci Rep 11, 22302 (2021). https://doi.org/10.1038/s41598-021-01636-1 ;
ii) McKnight, D. T., Huerlimann, R., Bower, D. S., Schwarzkopf, L., Alford, R. A., & Zenger, K. R. (2019). Methods for normalizing microbiome data: an ecological perspective. Methods in Ecology and Evolution, 10(3), 389-400.),
the comparison of NGS profiles from libraries with different sizes may lead to biases in the alpha and beta biodiversity indicators. The authors need to be more specific about how they have taken this issue into account, and why they have or have not chosen to apply a normalization / rarefaction step, and under what conditions.

Validity of the findings

The authors indicate they want to present this study as an initial case study. The term “case study” should be mentioned in both abstract and title to avoid wrong expectations about the manuscript content. The work presented in this article is indeed a very interesting new approach, but the title “patterns of colonization and potential sampling bias” suggest a larger pool of sampled houses as there is a lot of potential microbiome variation between dwellings.

Additional comments

The new approach presented in this study is promising and very interesting. The study of the microbiome inside building materials and the impact of sampling plan provide useful information for further analyses. My main concern is still the lack of sampling building replicates, which decrease the potential generalization of these results to other sites and environmental contexts. But as a case study, this article could be relevant after answering the few remaining questions.

Reviewer 2 ·

Basic reporting

Upon this resubmission, Davies et al. have substantially improved the manuscript and thoroughly addressed reviewer comments. While I think the dataset this publication is based on is still quite limited in scope (one building at one time point) and measurements of additional parameters like total biomass would strengthen the work, I think this manuscript is now largely suitable for publication. I commend the authors for their quick and relatively thorough work adding in taxonomic analyses and relevant statistics. The discussion section is also much improved, with more engagement with prior built environment literature.

Experimental design

no additional comments

Validity of the findings

New methods such as FEAST sourcetracking and multivariate statistics are appropriate and well described, including submission of relevant updated code.

Additional comments

I don't think "Group" is an effective legend title for the colors in supplemental figures 3 & 4, please change to "Predicted source" or similar.

Reviewer 3 ·

Basic reporting

The updates that the authors added were very useful across the manuscript. I appreciate the time that they have put into updating things, particularly in expanding the discussion. The citations added to the introduction also help with putting the study in context with others that have considered buildings as ecosystems, while making more clear the gap their study is addressing.

The addition of the discussion of the specific taxa greatly enhances the paper. I would suggest that the authors consider whether it should come after their original initial paragraph. It would make more sense to me to first present the alpha diversity, but this is a presentation choice that I leave to them.

I appreciate that the authors highlighted which specific walls were sampled in Figure 1. This is very helpful. However, related to the environmental conditions added, it is challenging reading the numbers on the sides of the building. Would it be possible to add those without the perspective?

I also found some additional typos in the new text including the following:
- line 14: It should be "communities" and not "community".
- line 55: There is a missing to in the clause "but due to air flow". It might be helpful to rephrase that part entirely to be "but humidity can be found in these spaces due to air flow."
- line 84: I think it would be clearer to use "insufficient" rather than "limited" here.
- line 88-89: I would replace "to make a contrast between" with "to compare the"
- line 128: cutadapt was misspelled
- line 135: Why is decontam italicized?
- line 170: I find the clause "To examine the likely most important taxa..." awkward. Maybe it could be deleted?
- line 194: I would change “to building” to be “from building”
- line 203: There is a missing “to” before “human as a source”
I would recommend generally reading through the manuscript again because I didn't include all here.

Experimental design

Overall, the updates that the authors added in regards to my comments are thorough and address my concerns.

I appreciate the extra text added to more fully discuss the impact of humidity on the microbes. My remaining comment is related to the language added at line 247, which I find confusing. Does the intensity of direct sunlight matter or are the microbial changes simply from humidity? If humidity is the key parameter, perhaps the authors do not need to mention sunlight at all?

Validity of the findings

My comments have been dealt with for which I thank the authors.

Additional comments

Thank you for addressing these additional comments. Overall, my concerns have been addressed; however, I have a two follow up points:

1. I’m still finding the title a bit confusing. Would it be possible to use some other word beyond “building elements” in the title? I wouldn’t have any idea what this means without reading the paper. Maybe something like “building envelope” instead?

2. At lines 46-52, I appreciate the updates the authors made to the text. I would recommend further editing it to the following: "but these samples cannot separate the microbial contributions of the structural aspects of the building (e.g., the building envelope) from those derived from the building occupants (e.g. hair or skin) (Cao et al., 2021) or the surrounding environment (e.g. soil, pollen grains) (Barberán et al., 2015a)." I think that this would better reflect that the issue with previous swabbing efforts (in my mind) isn't that they can't detect the effect from these structural taxa, but that they can't be identified as such.

---

## Round 0.3 · accepted · Accept

I have taken over handling the submission because the original Academic Editor is not available.

The authors have addressed all of the previous comments from reviewers, and while they have not performed rarefaction there has been an effort to remove bias introduced via sequencing depth. As such, I don't feel the need to send this manuscript out to reviewers again and feel that it is ready for publication.